

# Vocal constraints on song amplitude in star finches *Bathilda ruficauda*

Hana Goto[1], Masayo Soma[2], Ayumi Mizuno[3,4] and Henrik Brumm[5]

[1] Graduate School of Life Science, Hokkaido University, Sapporo, Hokkaido, Japan
[2] Faculty of Science, Hokkaido University, Sapporo, Hokkaido, Japan
[3] Faculty of Science, University of Alberta, Edmonton, Alberta, Canada
[4] Faculty of Science, University of New South Wales, Sydney, New South Wales, Australia
[5] Animal Communication & Urban Ecology Group, Max Planck Institute for Biological Intelligence, Seewiesen, Germany

## ABSTRACT

Given the two main functions of birdsong, mate attraction and territory defence, song amplitude is crucial for communication because it determines the communication range and it can also carry information. To understand the evolution of signals, it is helpful to consider the constraints on signal production, as physical constraints set the limits for signal plasticity and how signals can respond to selection. Previous work on the constraints of song amplitude was restricted to species that use loud vocalisations for long-distance communication. However, the low song amplitudes of some non-territorial species may hint at unknown limitations considering that females may prefer loud song. The star finch (*Bathilda ruficauda*) is one such species. In this study, we recorded the songs of male star finches in the laboratory and investigated the relationship between vocal amplitude and other acoustic parameters of their song syllables. We found that vocal amplitude was linked to the phonetic structure of the syllables. More complex sounds (measured as higher syllable bandwidth and higher Wiener entropy) were produced at lower amplitudes than less complex sounds. To the best of our knowledge, this is the first report of a trade-off between song complexity and vocal amplitude. More complex song syllables possibly require a more intricate control of the syrinx and vocal tract, which might only be possible at lower amplitudes. We speculate that if female star finches prefer complex songs, this would cause males to sing quietly, as they cannot produce complex syllables with high amplitudes. We also provided a phonetogram (vocal range profile) for the star finch, indicating a positive correlation between amplitude and peak frequency. This finding corroborates earlier studies on species that use loud vocalisations for long-range communication, which, like the star finches in our study, were also unable to produce low frequencies at high amplitudes. This suggests that the frequency-amplitude correlation is a more general phenomenon in bird vocalisations, independent of the overall source amplitude of a species. While the evolution of birdsong is often viewed as constrained by neural costs and body size, or selective pressures from predators and social aggression, our results emphasise the importance of understanding phonetic features as well. At the same time, our study fills an important gap on non-territorial species that produce soft songs. The absence of the ecological demands for long-distance signalling has probably led birds to use soft yet complex songs that function within the pair bond, as we report here for star finches.

Corresponding author
Masayo Soma,
masayo.soma@sci.hokudai.ac.jp

## INTRODUCTION

Birdsong is one of the most widely studied models of acoustic communication. As it mainly functions in mate attraction and territorial defence (*Catchpole & Slater, 2008*), its evolution is linked to sexual selection (*Searcy & Andersson, 1986*). While the spectral and temporal parameters of birdsong, such as pitch, repertoire size, trill rate, and song timing, have been well studied (*Catchpole & Slater, 2008*), much less is known about the amplitude of song due to the technical challenges of accurately measuring it, such as controlling the distance between the microphone and the signaller, the orientation of the signaller, and the environmental acoustics (*Brumm & Zollinger, 2011*; *Zollinger & Brumm, 2015*).

Signal amplitude is critical in acoustic communication because it determines the communication range: louder signals can be transmitted over longer distances and the signal-to-noise ratio is crucial for receivers to detect and recognise signals (*Klump, 1996*; *Brumm, 2013*). In addition, the amplitude of a signal can also carry information that receivers use to make decisions (*Zollinger & Brumm, 2015*). Supporting the idea that song amplitude is sexually selected (*Gil & Gahr, 2002*), female red-winged blackbirds (*Agelaius phoeniceus*) and zebra finches (*Taeniopygia castanotis*) preferred relatively louder songs (*Searcy, 1996*; *Ritschard, Riebel & Brumm, 2010*). Song amplitude is also used in male-male competition, with louder songs triggering more aggressive reactions (*Espmark et al., 2010*; *Brumm & Ritschard, 2011*; *Ritschard et al., 2012*). However, amplitude variation between males remains mostly unexplored, except for the fact that song amplitude is largely independent of body size (*Brumm, 2009*).

When trying to understand the evolution of signals, it is useful to determine the constraints on signal production. In birdsongs, certain elements are more challenging to produce than others because of physical constraints (*Podos, 1997*; *Podos, Southall & Rossi-Santos, 2004*; *Suthers & Zollinger, 2004*; *Geberzahn & Aubin, 2014*; *Jakobsen et al., 2021*), causing parameter correlations or trade-offs between amplitude and other acoustic parameters, such as frequency or duration (*Osmanski & Dooling, 2009*; *Ritschard & Brumm, 2011*). One possible mechanical constraint is the phonetic properties of the vocal tract, which tends to amplify high-frequency sounds more efficiently (*Bradbury & Vehrencamp, 2011*). Therefore, it is probably challenging for birds to produce low-frequency sounds at high amplitudes. Amplitude and frequency are also associated because both parameters can be directly coupled during phonation: in the absence of any countermeasures by the syringeal muscles, an increase in air sac pressure required to increase vocal amplitude will drive the system harder and increase the tension of the vibratory masses in the syrinx, which in turn leads to higher fundamental frequencies (*Elemans et al., 2015*). This coupling is evident in the vocalisations of non-songbirds (*Beckers, Suthers & Ten Cate, 2003*; *Osmanski & Dooling, 2009*; *Schuster et al., 2012*), whereas songbirds, with their more intricate control of the syrinx, can uncouple amplitude and frequency during phonation and hence stay on pitch while increasing vocal amplitude

(*Zollinger et al., 2017*). Nevertheless, the vocal tract resonances will enhance high frequencies more effectively also in songbirds.

A phonetogram (vocal range profile) is a diagnostic plot of vocal amplitudes across frequencies (*Titze, 1992*). Such a plot for blackbird (*Turdus merula*) song demonstrates that peak frequency and amplitude are indeed positively correlated in this species (*Nemeth et al., 2013*). In particular, while blackbirds can produce high frequencies at both high and low amplitudes, low frequencies are produced at low amplitudes only (*Nemeth et al., 2013*).

Another possible constraint on the production of vocal amplitude in birds is the time required to reach the necessary air sac pressure and airflow speed to produce loud sounds (*Suthers, Goller & Wild, 2002*; *Goller, Mallinckrodt & Torti, 2004*; *Plummer & Goller, 2008*). Positive correlations between amplitude and duration have been found in zebra finch songs (*Ritschard & Brumm, 2011*) and budgerigar (*Melopsittacus undulatus*) calls (*Osmanski & Dooling, 2009*). In contrast, negative correlations between duration and amplitude have been reported for towhees calls, *Pipilo erythrophthalmus*, (*Nelson, 2000*) and for extremely loud vocalisations of white bellbirds, *Procnias albus*, and screaming pihas, *Lipaugus vociferans*, (*Podos & Cohn-Haft, 2019*), suggesting a different trade-off, perhaps between upregulation of respiratory air flow for the high-amplitude vocalisation and, on the other hand, depletion of respiratory tidal volumes (*Podos & Cohn-Haft, 2019*).

While some studies have looked at the relationship between frequency and amplitude in birdsong, only very few provide comprehensive measurements within a species and allow for strong inference across individuals (*Riede et al., 2006*; *Francis, Ortega & Cruz, 2011*; *Cardoso & Atwell, 2011*; *Gough, Mennill & Nol, 2014*). The only published phonetogram of a bird, covering the full vocal range, comes from a territorial species that sings loud songs that are used for long-range communication (*Nemeth et al., 2013*). However, some species sing at much lower amplitudes, especially those that do not defend territories with their songs (*Goodwin, 1982*). How is amplitude constrained in these species? To answer this question, we examined the songs of male star finches (*Bathilda ruficauda*), an estrildid finch species. Among estrildid finches that generally lack a clear territory (*Goodwin, 1982*) and are characterised by close-distance communication, such as singing (*Loning, Griffith & Naguib, 2022*) and dancing (*Ota, Gahr & Soma, 2015*; *Soma & Iwama, 2017*; *Gomes et al., 2017*), star finches have remarkably soft songs that were described as audible to human ears only within a few metres (*Goodwin, 1982*). Indeed, we found that the source level of star finch song is, on average, as low as 43 dB(A) SPL at 1 m (H. Goto, 2023, personal observation). Star finch song typically consists of simple introductory notes, followed by motifs composed of several syllable types that can have complex acoustic structures (Fig. 1, Fig. S1).

Here, we recorded star finches in the laboratory to investigate the relationship between amplitude and other acoustic parameters. We predicted parameter correlations: a positive correlation between amplitude and peak frequency of a syllable based on the phonetic properties of vocal tract, and a positive correlation between amplitude and duration of a syllable based on the time required to reach the necessary pressure or flow to produce a loud vocalisation. Given the soft song of the species and its complex song syllables, we also hypothesised a trade-off between amplitude and acoustic complexity of a syllable because

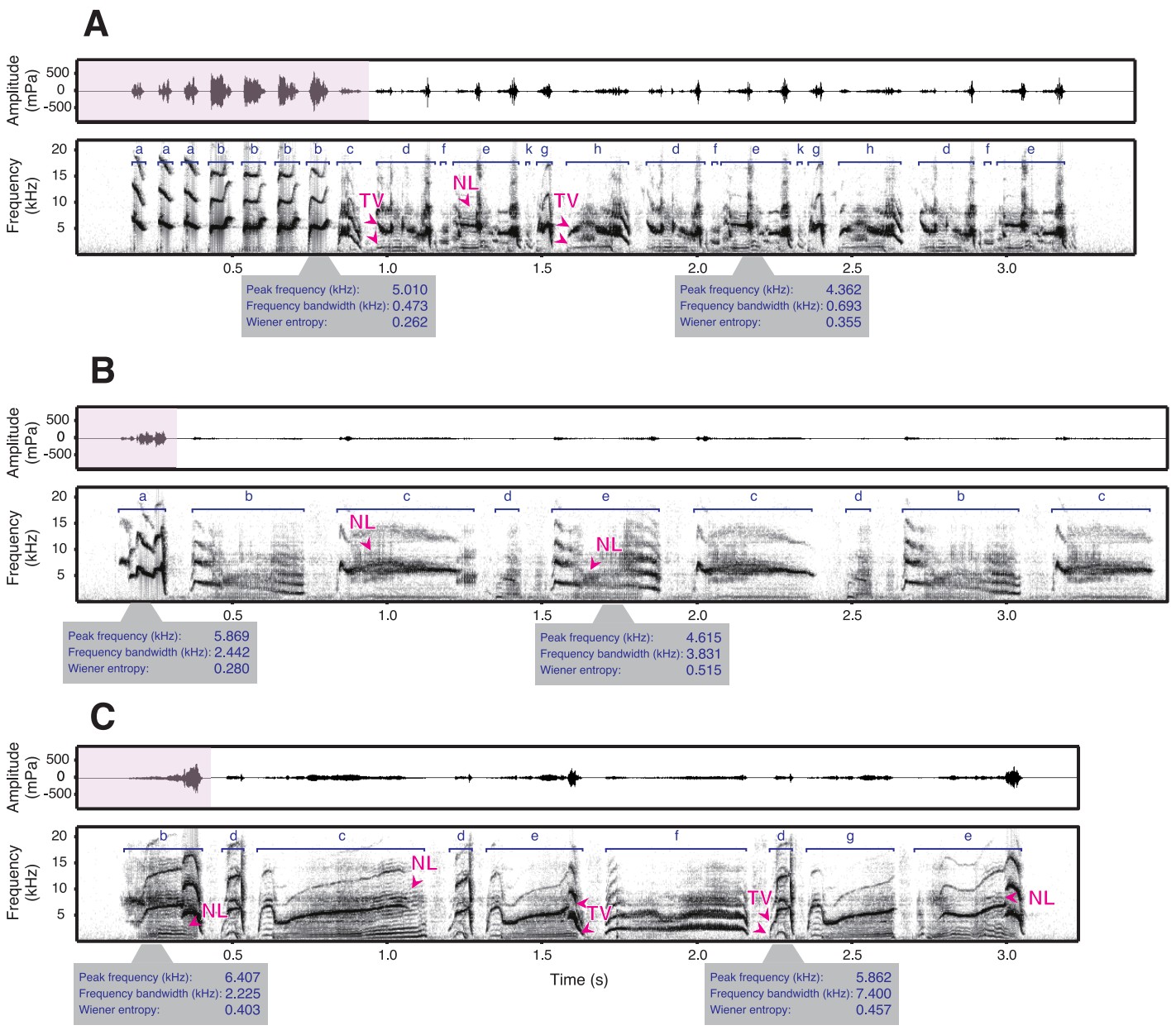

**Figure 1 Spectrograms and waveforms of star finch songs.** The song types of three exemplary males (A: SF0002, B: SF0031, and C: SF0037) are shown (see Fig. S1 for songs of additional males). Star finch song typically consists of one or several comparably loud introductory syllables (pink parts of the waveforms), followed by softer syllables of the main phrase. The syllable types are shown in dark blue. The syllables of the main phrase can have many acoustically complex features such as frequency modulation, suspected two voices (TV), and nonlinear phenomena (NL) (the latter two indicated with arrows in dark pink, see Methods for details). Two representative syllables (*i.e.*, one introductory and the other main phrase syllables) per male are shown with acoustic parameters. In addition, star finches often produce very soft vocalisations between syllables, making it difficult to separate syllables automatically (see Methods for how sequences were segmented).

complex syllables that are more difficult to produce may only be sung accurately at lower amplitudes (*Suthers & Zollinger, 2004*), and broadband syllables may not completely match the amplified frequency of the vocal tract. Complexity was measured as frequency bandwidth and Wiener entropy. Frequency bandwidth is a parameter of frequency

modulation, and higher frequency bandwidth indicates higher relative complexity. Wiener entropy is a measure of the width and uniformity of the power spectrum. It is zero for simple pure tones, while higher values indicate higher relative complexity. For instance, the use of dual sound sources (two voices) and nonlinear phenomena, *i.e.*, irregular sounds, such as biphonation, sidebands, deterministic chaos, or subharmonics (*Zollinger, Riede & Suthers, 2008*; *Amador, Mindlin & Elemans, 2025*), increase Wiener entropy (*Anikin & Herbst, 2025*).

## MATERIALS AND METHODS

### Subjects

Fifteen adult male star finches were sourced from local breeders *via* the pet shop, Limited Company Koizumi, Sapporo, Japan. Except for the sound recording period, the birds were kept in group living cages: 14 birds were kept in unisex cages (39.5 cm 32.0 cm 63.0 cm high), with four or five birds per cage, and one male (SF0037) was kept with a female in a cage (35.0 cm 28.5 cm 39.5 cm high). The cages were kept on a 12:12 h light: dark schedule (lights on 08:00–20:00) at approximately 25–26 °C and 50–60% humidity. The finches were fed *ad libitum* on a diet of mixed seed supplemented with egg yolk-coated millets, finch pellets, and fresh green foods. Fresh drinking water and oyster-shell grit were provided *ad libitum*. All birds were kept and treated in accordance with the Japanese Act on Welfare and Management of Animals. Our animal use protocol was approved by the Institutional Animal Care and Use Committee of Hokkaido University (Approval No. 22-0052). The study was purely observational and did not involve any invasive procedures or interventions.

### Song recording

All recordings were made between November 2023 and March 2024. Each male was placed in a cage (31.0 cm 25.5 cm 42.0 cm high) in a soundproof chamber for about 24 h (from around 19:00 to 19:00 the next day). The bird was moved back to the original cage after the recording. Each cage was equipped with a single perch in the centre and water and food cups. To minimise the measurement error of the song amplitude, we paid particular attention to (1) the social context and background noise, (2) the direction of the bird's head, and (3) the distance between the bird and the microphone.

(1) Because songbirds are known to vary their song amplitude depending on the level of background noise (*Brumm & Zollinger, 2011*) or social context (*Cynx & Gell, 2004*; *Brumm & Slater, 2006*), the birds were recorded individually in a soundproof chamber. This setup allowed us to keep these environmental variables constant across and within all recorded males (c.f. *Goto et al., 2023*).

(2) The recorded song amplitude could vary due to the lateral head movements of the birds because birdsong shows a directional sound radiation pattern (*Brumm, 2002*). To control this, we placed the internal microphones of a recorder (PMD661; Marantz, Cumberland, USA) above the centre of the perch, approximately 11 cm from the bird's

head. Digital audio recordings were made continuously at a sampling rate of 44.1 kHz with 16-bit accuracy.

(3) For analyses, we used songs produced only at the perch to control the distance between the microphone and the singing bird. The location of the bird was determined by listening to the recordings. When the bird was on the perch, he produced scratching sounds on the wood, which were clearly different from the sounds produced on the newspaper sheet at the bottom of the cage or on the plastic feeding cups.

To calibrate the sound amplitude measurements, we recorded a sequence of six sine tones at 6.0 kHz, increasing in 5-dB steps as a reference signal for the analyses. These sine tones were played through the internal speaker of another Marantz PMD661 from a fixed location in the soundproof chamber. Then, the first recorder was replaced with a sound level meter (Digital Sound Level Meter SL8850, AS ONE, Osaka, Japan), and the tone sequence was played again to measure the amplitudes of the calibration signals.

## Song analysis

The song analyses were conducted using the Avisoft-SASLab Pro 5.3.2-14 software (Avisoft Bioacoustics, Berlin, Germany). We calibrated the amplitude measurements with the sine tones, using the "SPL with reference sound" function of the software. For the analyses, we chose 20 songs per individual produced between 8:00 and 14:00. This time window was chosen to ensure a sufficient number of songs produced while perched, accounting for variations in singing activity across individuals and times of day. We aimed to select songs as evenly distributed as possible across the time period. For each bird, we selected the songs with the longest sequences to consider as many syllable types as possible. Prior to the analyses, the song recordings were high-pass filtered (cut-off frequency: 0.2 kHz; Hamming window) to remove low-frequency noise.

As a first step in the song analyses, we labelled the syllables of the songs. Star finches often use potential two voices or nonlinear phenomena in their song elements and have soft sounds between their syllables (Fig. 1, Fig. S1). It is not possible to distinguish between two-voiced sounds and biphonations—a specific type of nonlinear phenomenon—based on sound recordings alone (*Zollinger, Riede & Suthers, 2008*; *Amador, Mindlin & Elemans, 2025*). In Fig. 1, we use the term "two voices" to describe cases where the observed pattern could be attributed to either two voices or biphonation. These unique song characteristics of this species often make it impossible to split syllables into elements. Syllables were first automatically detected using an amplitude threshold with a minimum syllable duration of 5 ms and a minimum inter-syllable gap of 5 ms. Then, the labels were visually checked. We sometimes modified the length of the labels so that all syllables of the same type were labelled in the same manner for all songs. This manual editing was done because automatic detection occasionally labelled non-vocal noises or sometimes labelled two successive syllables as one syllable due to low-amplitude sounds between the syllables.

First, we categorised the syllables into syllable types. For this we used the automatic classification of labelled sections in Avisoft, based on manually annotated labels as a reference. Then syllable classifications were verified by hand because in some cases the

automatic function did not differentiate similar syllable types or failed to recognise syllables. Star finch songs consist of repetitions of a basic motif that form a continuous phrase (*Goodwin, 1982*, Fig. 1). Syllables that are not contained in the motif and typically produced at the beginning of a phrase were defined as introductory syllables, following the terminology used for zebra finch song (*Kalra et al., 2021*). One bird (SF0026) did not repeat the motif in the song. For this bird, we defined the first syllable in a song bout as introductory syllable.

To investigate whether syllable amplitudes were affected by potential physical constraints and acoustic complexity, we measured amplitude and other four acoustic parameters of the labelled syllables. The amplitude was computed as the root mean square value of the entire syllable from the envelope (dB SPL, re. 20 μPa). The peak frequency, syllable duration, Wiener entropy, and frequency bandwidth were computed from the spectrograms (FFT length = 256, Hamming window) using the automatic measurement function of Avisoft.

### Statistical analysis

We analysed the relationships between the amplitude and the measured acoustic parameters using a linear mixed model built with the lmer function from the package lme4 (v. 1.1-37, *Bates et al., 2015*) in R (v. 4.2.2, *R Core Team, 2024*). The model was fit by maximising the restricted log-likelihood. The four acoustic parameters and the syllable categories (introductory or main phrase) were considered as fixed effects. The random effects structure in our model had a three-level hierarchical nesting where syllable type ($N = 126$) was nested within song ID ($N = 300$), and song ID is nested within bird ID ($N = 20$). This accounts for variation at the individual level (bird ID), between different song bouts produced by the same individual (song ID), and among different syllable types within each song (syllable type). Heteroskedasticity and distribution normality were checked visually and using the bptest function in lmtest package (v. 0.9-40, *Hothorn et al., 2022*). To further assess the influence of the random effects, models without random effects (bird IDs, song ID, syllable types) were built and compared to models with full random effects, using anova function in R.

In addition, to assess the variance explained by fixed effects and the variance explained by both fixed and random effects (*Nakagawa & Schielzeth, 2013*; *Johnson, 2014*), we calculated marginal $R^2$ and conditional $R^2$ using r.squaredGLMM function in MuMIn package (v. 1.40.4, *Bartoń, 2025*).

To evaluate the effect of the different housing environment on SF0037, we ran the same statistical model without this individual, but the results remained the same.

## RESULTS

All males sang in the soundproof chamber. Overall, we analysed 5,539 syllables from 20 songs of each of the 15 birds. Between males, the maximum number of syllables per song varied from eight to 55. Star finches have only one song type, and the syllable order was stereotyped within individual males (Supplemental Data: syllable type order).

**Table 1 The results produced by a linear mixed model evaluating the relationships between amplitude and the measured acoustic parameters, and results of model comparison.**

| Fixed effect | Estimate | SE | DF | t | p |
|---|---|---|---|---|---|
| (Intercept) | 69.53 | 1.33 | | | |
| Peak frequency (kHz) | 1.60 | 0.06 | 5,519.18 | 25.23 | <0.001 |
| Duration (s) | 5.75 | 0.74 | 3,241.84 | 7.82 | <0.001 |
| Frequency bandwidth (kHz) | −0.50 | 0.03 | 4,471.60 | −15.32 | <0.001 |
| Wiener entropy | −38.17 | 1.22 | 5,056.59 | −31.42 | <0.001 |
| Introductory syllable | 3.75 | 0.26 | 2,267.77 | 14.62 | <0.001 |
| **Random effect** | | $\chi^2$ | | | p |
| Bird ID | | 6,308.7 | | | <0.001 |
| Song ID | | 0 | | | >0.999 |
| Syllable type | | 588.46 | | | <0.001 |

First, we investigated the relationship between the amplitude and acoustic parameters related to potential vocal constraints in a syllable. Peak frequency had a strong effect on the amplitude of the syllables ($p < 0.0001$; Table 1), with higher-pitched syllables being produced at higher amplitudes (Fig. 2A). This relationship was observed in almost all individuals (14 of 15 males; Fig. S2). Likewise, the tested birds also produced longer syllables at higher amplitudes ($p < 0.0001$; Table 1; Fig. 2B).

In addition, we investigated whether there is a trade-off between syllable amplitude and the acoustic complexity of a syllable. More complex syllables, those with larger frequency bandwidths or higher entropy, were sung at lower amplitudes than less complex syllables (frequency bandwidth: $p < 0.0001$; Table 1; Fig. 2C; entropy: $p < 0.0001$; Table 1; Fig. 2D). We also found that introductory syllables in the beginning of a song were considerably louder than the motif repetitions of a song's main phrase ($p < 0.0001$; Table 1, Fig. 2E). To account for the effect of SF0020, which produced quiet and broadband syllables (Fig. 2C), we ran the same statistical model without this individual, but the results remained unchanged.

The model comparisons with different random effects indicated that individual males and syllable types differed significantly in song amplitude (effect of bird ID: $p < 0.001$; effect of syllable type: $p < 0.001$; Table 1). Within males, however, the different song renditions did not differ significantly in their amplitudes (effect of song ID: $p > 0.999$; Table 1).

The marginal $R^2$ and conditional $R^2$ of the full model were 0.427 and 0.970 respectively.

## DISCUSSION

To understand possible constraints on birdsong, we investigated relationships between amplitude and other acoustic parameters of syllables in star finch songs. We found that vocal amplitude varied strongly with the phonetic structure of the syllables (Fig. 2, Table 1). Supporting our prediction of possible physical constraints, we observed a positive correlation between amplitude and peak frequency as well as amplitude and syllable

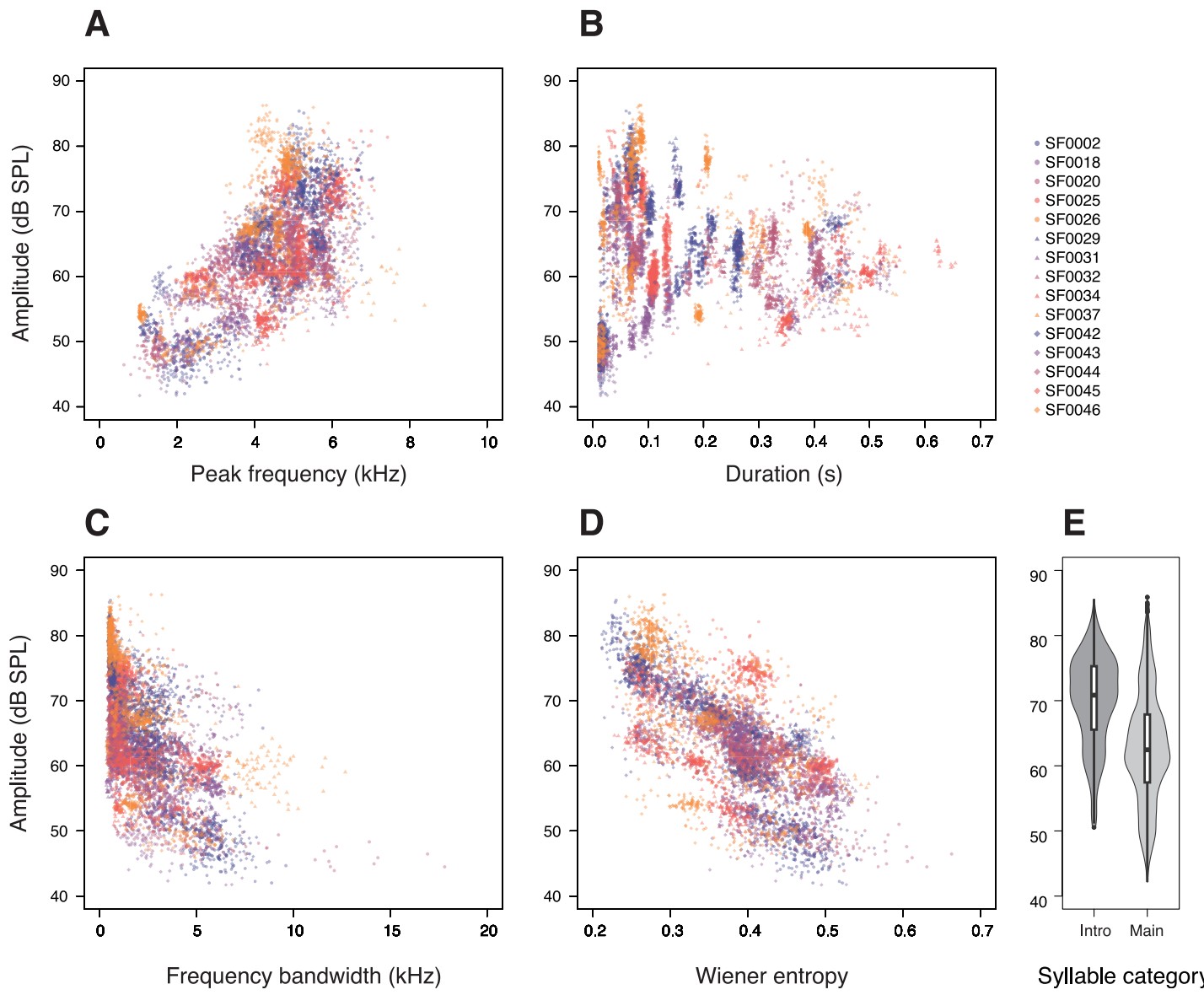

**Figure 2 Relationships between amplitude and (A) peak frequency, (B) duration, (C) frequency bandwidth, (D) Wiener entropy, and (E) syllable categories in star finch songs.** Each dot represents a single syllable, colours and shapes correspond to individual birds. The amplitude values were measured at 11 cm above the birds. (E) "Intro" denotes introductory syllables and "Main" denotes syllables of the main phrase.

duration (Table 1). Most interestingly, we discovered negative relationships between the amplitude and acoustic parameters related to acoustic complexity (Table 1), as predicted. To the best of our knowledge, this is the first report of a trade-off between vocal complexity and amplitude in birdsong.

The positive correlation between the amplitude and peak frequency that we identified (Fig. 2, Table 1) can be accounted for by the potential physical constraint based on the phonetic properties of the vocal tract. Such parameter correlation is typically observed in
human vocalisations (*Titze, 1992*), as well as bird calls (*Nelson, 2000*; *Osmanski & Dooling, 2009*; *Dorado-Correa, Zollinger & Brumm, 2018*) and songs (*Ritschard & Brumm, 2011*; *Nemeth et al., 2013*). Compared with previous bird studies, our study involved a larger number of subjects and encompassed all song syllable types in an individual's repertoire. Thus, we can provide the most comprehensive phonetogram of a bird to date. Given that the vocal range profile of star finches is similar to that of blackbirds (*Nemeth et al., 2013*), we can conclude that the difficulty in producing low-pitched sounds at high amplitudes is not restricted to territorial species that sing very loudly. This relationship appears to be a more general phenomenon in birds.

We also found that syllable amplitude varied with the syllable duration. The amplitude of bird vocalisations is related to air sac pressure and airflow speed (*Suthers, Goller & Wild, 2002*; *Goller, Mallinckrodt & Torti, 2004*; *Plummer & Goller, 2008*). Therefore, the time required to reach the necessary pressure or flow to produce a loud vocalisation should affect the amplitude. Our data support earlier studies that reported positive correlations between amplitude and duration in zebra finch song elements (*Ritschard & Brumm, 2011*) and budgerigar calls (*Osmanski & Dooling, 2009*).

In addition to these parameter correlations, our findings indicate a trade-off between song amplitude and acoustic complexity, possibly caused by physical constraints. More complex syllables may require more intricate coordination between the syrinx and the respiratory muscles, which might only be possible at lower amplitudes (*Suthers & Zollinger, 2004*). This could also apply to other forms of vocal complexity that were not explicitly considered in our study, such as two voices or nonlinear phenomena. However, the precise proximate mechanism underlying this trade-off remains unclear. In any case, the observed frequency constraint on song amplitude could contribute to the amplitude-complexity trade-off because birds might produce loud songs only within the restricted range of frequencies that are amplified most effectively by their vocal tracts (*Nemeth et al., 2013*; *Wiley, 2015*). In other words, the frequency limitation of loud vocalisations eventually reduces the frequency variability at high amplitudes.

Regarding the ultimate causes, we speculate that female choice and social factors may have driven the evolution of soft songs in star finches and other estrildid finches. If females preferred complex songs, this would drive males to sing softly because they cannot produce complex syllables at high amplitudes. As star finches use their songs primarily during courtship (*Goodwin, 1982*), they do not need to transmit far; therefore, unlike territorial birds, star finch males can maximise complexity at the expense of amplitude. In zebra finches, several studies have suggested that their song is used for individual identification (*Loning, Griffith & Naguib, 2024*; *Bulla & Forstmeier, 2024*) rather than an indicator of male quality because female mate preference is independent of the song (*Forstmeier & Birkhead, 2004*; *Wang et al., 2021*). This might also be the case in star finches. However, our results do not allow any speculation about whether or not song complexity of star finch song is an identity signal or sexually selected trait.

Since the complex syllables are low in amplitude, they will not transmit far. In addition, more complex, broadband syllables will be affected more heavily by degradation during signal transmission, reducing their active space even further. Perhaps the

comparably loud introductory notes are used as an alerting signal to draw potential receivers closer so that they can hear the complex syllables that the males are unable to sing loudly.

Overall, our results suggest several phonological features that correlate with song amplitude, some of which can be taken as potential constraints. It is important to note that this cannot be separated from a potential influence of song learning. In zebra finches, the amplitude of tutee song elements was strongly correlated with that of tutor elements, which may be the result of vocal learning or simply reflect phonetic constraints (*Ritschard & Brumm, 2011*). However, considering that similar relationships between vocal pitch and amplitude have been observed across different species, including those that do not learn their vocalisations (*Beckers, Suthers & Ten Cate, 2003*; *Schuster et al., 2012*; *Dorado-Correa, Zollinger & Brumm, 2018*), we believe that our findings might be universally applicable to many bird species.

While the star finches in this study were recorded in isolation, future research exploring the effects of social context on song amplitude would be interesting. For instance, zebra finches regulate song amplitude depending on visual or auditory contact with either sex (*Cynx & Gell, 2004*; *Brumm & Slater, 2006*). Our findings on the trade-off between acoustic complexity and song amplitude may suggest that increases in amplitude may vary depending on syllable complexity, leading to different amounts of amplitude gain in different syllable types (c.f. *Brumm & Todt, 2002*).

## CONCLUSIONS

Limitations of birdsong have often been discussed in terms of costs from social aggression, predation, and neural costs, and constraints from body size (*Gil & Gahr, 2002*; *Mikula et al., 2021*). Here, we highlight the importance of phonetic constraints as well, which refer to limitations related to the mechanics of sound production. While our correlative data do not indicate performance limits in the relationships between amplitude and frequency or duration, they raise the possibility of trade-off between vocal amplitude and acoustic structural complexity. It would be interesting to explore these relationships in situations that require high vocal performance, such as singing to attract a distant mate or in noisy environments. If the observed gaps in parameters spaces indeed reflect vocal production constraints, birds should be unable to produce sounds in these areas. Understanding the physical boundaries of song performance are essential, because they limit how song can respond to selection. At the same time, our study fills an important gap regarding non-territorial species that sing soft songs. It is likely that the absence of ecological demands for long-distance signal transmission has led to the evolution of soft yet complex syllables, which function in short-range contexts, such as within pairs or social groupings.

## ACKNOWLEDGEMENTS

We thank the members of the laboratory in Hokkaido University for help with animal care. We also thank Devraj Singh and three anonymous reviewers for the time and effort they devoted to help us improve the manuscript.

### Funding
This work was supported by JSPS KAKENHI Grant Number 24KJ0314, 24K09553. The funders had no role in study design, data collection and analysis, decision to publish, or preparation of the manuscript.

### Grant Disclosures
The following grant information was disclosed by the authors:
JSPS KAKENHI: 24KJ0314 and 24K09553.

### Competing Interests
The authors declare that they have no competing interests.

### Author Contributions
- Hana Goto conceived and designed the experiments, performed the experiments, analyzed the data, prepared figures and/or tables, authored or reviewed drafts of the article, and approved the final draft.
- Masayo Soma conceived and designed the experiments, analyzed the data, prepared figures and/or tables, authored or reviewed drafts of the article, and approved the final draft.
- Ayumi Mizuno conceived and designed the experiments, authored or reviewed drafts of the article, and approved the final draft.
- Henrik Brumm conceived and designed the experiments, authored or reviewed drafts of the article, and approved the final draft.

### Animal Ethics
The following information was supplied relating to ethical approvals (*i.e.*, approving body and any reference numbers):

All animal experiments were performed in accordance with the Japanese Act on Welfare and Management of Animals. Our animal use protocol was approved by the Institutional Animal Care and Use Committee of Hokkaido University (Approval No. 22-0052).

### Data Availability
The raw measurements are available in the Supplemental Files.

### Supplemental Information
Supplemental information for this article can be found online at http://dx.doi.org/10.7717/peerj.19705#supplemental-information.

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
