# Peer review of "Vocal constraints on song amplitude in star finches Bathilda ruficauda"

_PeerJ, doi:10.7717/peerj.19705_

## Round 0.1 · original submission · Major Revisions

· Academic Editor

Major Revisions

This manuscript has received a very thorough review with helpful comments. Please address all the comments made on clarity in the Introduction and hypothesis. Please address comments on giving more clarity on the conceptual basis for the idea that singing at higher amplitudes should limit birds’ capacities to sing complex songs. Address carefully comments made in Experimental design, data analysis, and interpretation of the data.

In the end, the discussion needs more clarity.

We look forward to reviewing this manuscript after all the comments have been addressed.

Reviewer 1 ·

Basic reporting

The text is generally clear and unambiguous, and well written. That said, I have a few minor comments on grammar, syntax, etc. :

- Line 62: Note correct genus spelling is "Taeniopygia"; missing initial 'i' in text
- Line 83: suggest creating a new paragraph at “A phonetogram[...]” to more conceptually separate these ideas.
- Line 125 and 127: "ad libitum" should be italicized in each location
- Line 136: “19:00 next day” should be “19:00 the next day”
- Line 160: “amplitude” should be “amplitudes”
- Line 201: “song syntax of the songs” should either be “song syntax” or “syntax of the songs”
- Line 231: Fig. 1 should be Fig. 2
- Line 242: “airsac” should be “air sac”, while "air flow" should be "airflow"
- Line 265: "estrildids" is cumbersome and should be either "estrildid finches" or "other members of the family estrildidae", or similar.
- Lines 327-328: Citation (Lampe et al., 2010) out of order in reference list; but, also appears to not be used in main text. Please remove if in error, else return the citation to the main text where it should appear.

The literature references are well-sourced, and the background is generally sufficient to provide context for this addition to the field. That said, I also have some minor comments on the background source material and its use:

- Lines 31-33: the implication is that higher syllable bandwidth and higher Wiener entropy equal “more complex” sounds. These two metrics are not direct measures of complexity, which is a subjective ascription - the strength is in the relational nature, i.e., one syllable might be considered more complex because it has a higher entropy measurement than a different syllable. This implication should be made explicit for the lay reader with less familiarity of these measures, perhaps as simply as saying that “higher frequency bandwidth or Wiener entropy indicates higher relative complexity”. This point reverberates in lines 114-116.

- Lines 55-56: the framing would benefit from some further relatively short descriptions about what technical challenges have existed in accurately measuring amplitude, perhaps from the referenced citation.

- Lines 65-66: The reference found song amplitude to be mostly independent of body size, though did note an interesting significant negative correlation between body size and song amplitude during interactive singing in nightingales. This is an important clarification to be made, as the relationship exists during interactive singing, such as in a pair bond - which is a both a main facet of the author’s framing in this manuscript, and not a type of singing that was measured here, as the birds in this manuscript were recorded while singing alone. I suggest qualifying this statement (e.g., “song amplitude is [largely] independent of body size”), and further speculating on the interactive component of vocal production in the discussion (more in the final section).

- Lines 96-97: The citation does not include a phonetogram, as defined in the manuscript. Perhaps this is meant to be Nemeth et al., 2013, which is mentioned further above as a published phonetogram (lines 85-88). In addition, there have been many phonetograms published for birds (http://dx.doi.org/10.1073/pnas.0601262103, http://dx.doi.org/10.1676/13-088.1, https://dx.doi.org/10.1098/rsbl.2011.0359, https://doi.org/10.1016/j.anbehav.2011.07.018), though I wholeheartedly agree with the authors that Nemeth et al., 2013, has perhaps the most exhaustive published phonetogram for a species where we can be reasonably certain this is the vocal range profile within a species. I suggest qualifying this sentence to reflect the fact that, while many phonetograms have been published for birds, there are relatively few (perhaps two, once this paper is published) that are comprehensive within a species and allow for strong inference across individuals.

- Lines 251-252: The provided citation is not of the zebra finch, but the chaffinch.

The structure is easy to understand, and the figures are mostly without issue. That said, I have some concerns about the figure reporting:

- The description of syllable labeling is incredibly helpful, and shows laudable rigor. However, while the authors mention visual checking and manual editing, providing more discrete examples about syllable boundary definitions would be helpful. Adding syllable labeling on the song examples in Figure 1 would be extremely beneficial, and satisfy this point. Further labeling on the sonograms included in Supplementary Figure 1 would be additionally useful, but I leave that to the authors’ discretion.

- In both Fig. 1 and Fig. S1, please provide the numeric scale on every y- and x-axis. The legend (Amplitude, Frequency, etc.) need not be applied to every plot, but the numerics must be included for sake of reporting rigor. Otherwise, we must take the author's word they are plotted on the same scale without direct confirmation.

The aims and hypotheses are clear, and the results reported speak directly to these. Overall, this is a solid paper, and an excellent resource to the field.

Experimental design

The research reported here is certainly original, and it bears repeated a supremely necessary gap that needs filling. I appreciate immensely the deep description of the potential issues of measuring sound amplitude, and the authors’ explicit description of how they accounted for these potential issues, so as not to bias their data collection and analyses. The research question is targeted, well-defined, and the results directly address the question. Though the methods and investigation are relatively solid, I have some major concerns about the linear modeling and thus how the main results are reported, and some minor concerns about the metadata of the song files analyzed.

Major concerns:

- The justification for using a linear mixed model is sound. However, mentioning whether model assumptions (e.g., normality of residuals, homogeneity of variance) were checked, and how, would add rigor. A cursory analysis of the provided raw data used to generate the model indicates there may be some heteroskedasticity. In addition, it is not obvious that song_ID was nested within bird_ID in the model. If songs are unique to individuals, this hierarchical structure should be reflected in the random effects. The methods section does not clarify this relationship directly - this should be made explicit if the nested model was indeed used, and justified if not. In addition, it is not described which lme 'method' in the package is used - are the authors maximizing the restricted log-likelihood? Or are they maximizing the log-likelihood? This should be justified, and is necessary for one to replicate the results.

- While including syllable order as a fixed effect makes sense to some first approximation given its influence on amplitude, the description does not clarify how this variable is structured (e.g., continuous or categorical). Modeling it incorrectly might lead to spurious results. But more importantly, the inclusion of syllable order in the model may not be methodologically justified, given what appears to be high variability based on syllable_order number (more in the next section). I am concerned about the inclusion of this term in the model, and outside of further justification from the authors, I would recommend either its removal, or reannotating as a categorical variable (e.g., "introductory note" vs "main song syllable"), for a reanalysis.

- Line 123: which male was kept with a female? Were the song qualities recorded substantially different than the males kept in unisex housing? Given the known role of song in the pair bond, and the context-dependency of vocal features in many species, it would be useful to know whether the song recorded from a bird immediately after being removed from a potential mating context is different from those who were non-mating. In addition, it would be useful to anchor this knowledge to the results - is the single male who was kept with a female the outlier who did not produce higher-pitched syllables at higher amplitudes (lines 206-207)? Did this bird produce songs with relatively many or few syllables? Was this the bird who appears to be a potential outlier in the lower right quadrant of the bandwidth vs amplitude plot in Fig. 3a? Etc. At the very least, a control should be acknowledged to compare this single individual to the remainder of the individuals, given the extreme difference in environment, to ensure the resulting data is not unduly biasing the modeling and results - and remove him from analysis, if the data warrants.

Minor concerns:

- I was interested to see the songs analyzed came from across a long time period in the day (from immediately after lights on until about 6 hours later), yet analyzed 20 songs only across that time. There are well-documented changes in song across the day, particularly in the zebra finch and particularly in the first few hours of the light cycle (DOI:10.1523/JNEUROSCI.3387-05.2006). As such, many researchers tend not to analyze song in these first few hours of the day, due to the higher variability than later-day songs. The justification for this long time-period could be more explicit, which would strengthen the methodology. However, I suggest revisiting the metadata of the recording files, and reporting the relative time frames from which recorded songs were included in the analysis. If songs were recorded in a 6 hour window, but only songs within a 2 hour window were included in the analysis, this would further support the subsequent analyses as unbiased. To be clear, I do not believe the specific recording times need be worked into the model; this is simply an important control that should be mentioned if already done, and analyzed if not, for rigor's sake.

- While p-values are reported, including effect sizes in the main text (e.g., estimated coefficients from the mixed model, R-squared values for fixed effects) would provide a more complete picture of the strength of the relationships. For instance, the statement about the non-significant effect of syllable duration ("longer syllables also tended to be produced at higher amplitudes, but this difference was statistically insignificant"; lines 207-208) could be improved by providing the estimated effect size and confidence interval, in addition to the p-value, in the main text. This would allow the reader to assess the magnitude of the effect there, even if it wasn't statistically significant.

Validity of the findings

The underlying data have been provided and are well annotated, allowing my own cursory assessment of the dataset, which is greatly appreciated for evaluating the results reported. The data seem robust to some first approximation, though I have noted some issues with potential statistical soundness and control in the section prior. The conclusions are mostly well-stated, and for the most part related to the actual results reported. That said, I have some additional comments, and some prior bear repeating:

- Lines 201-202: The actual invariability of song syntax is never explicitly tested, yet stated as a result. In fact, this is counter to the sentence immediately before, which reports the “maximum number of syllables per song varied”, indicating there is, in fact, variability. When looking at the raw data provided, the statement in lines 201-202 would suggest that each entry for a single number in “syllable_order” should be incredible similar in spectral features reported - yet, this is not the case, and many syllables (by order number) show remarkable variability within bird, for many individuals and for many measures. Perhaps this is because the number of introductory notes vary across bouts for a single bird? At a minimum, this sentence should be removed; however, I suggest some consideration of actually plotting and analyzing this result, even as a supplemental figure, which would undoubtedly be a great resource to the community and aide in future replication or extension of these findings. Further, I suggest consideration of removal of this variable from the linear model, as it is not obvious that, e.g., syllable 5 is the same categorical variable across every bout of song within an individual (though see prior section and next comment for suggestions on how to modify this variable and maintain its inclusion in the model).

- In addition, I am worried about the inclusion of introductory notes in these analyses. As the authors state, star finch songs began with loud syllables and then decreased in amplitude as the song progressed (lines 214-215). What is really shown, at least in the few sonograms given, is that introductory notes tend to be higher amplitude than “true” song elements. In addition, this data is not plotted, and the conclusion is actually not supported - plotting syllable order vs for a few birds from the raw data file shows that, while this relationship is variable between birds, it is certainly non-monotonic. As mentioned in the comment above, and the prior section, I suggest either remodeling with introductory vs non-introductory note as a categorical variable, and/or plotting the actual song-level analysis, to support these claims and add further rigor to the results reported.

- The introduction and discussion lean quite a bit on referencing papers supporting the thesis that it is difficult for birds to sing lower frequency song elements at high amplitude. However, there are reports of birds who produce both short- and long-range songs with no apparent differences in frequencies (e.g., https://doi.org/10.2307/4089197, https://doi.org/10.1111/j.1439-0310.2008.01518.x). Some time, at least in the discussion, to address this counterpoint to the provided argument, and speculate on the current results in a comparison to these other reports, would strengthen the manuscript and its conclusions.

- Broadly, while I am wholly supportive of the supporting references in the introduction discussing the constraints of vocal production relative to the physical sound source and vocal tract filtering. However, I find some sentences missing citations (e.g., Lines 259-262). Please provide citations in the places where these supporting statements are made. In addition, I worry about the authors having too heavy a hand in calling to the fundamental properties of the vocal tract - often, they say because the vocal tract in some species can enhance high frequencies, and because peak frequency and amplitude are highly correlated, that means that the vocal tract must restrict frequency and, therefore, directly affect amplitude. These are false equivalencies, and I suggest making the language more speculative in the locations they initially appear.

- Lines 248-251: The statement of star finches often using syllables of a duration more than 200ms is actually not supported by the provided data, where the average syllable duration is closer to ~170ms. Even with the element fusing mentioned, if true (I cannot assess from the data provided), this would indicate that the average syllable length in the real behavior of the star finch might be even shorter. This is an important point, given that the following example is in contrast to the zebra finch (though incorrectly stated; the citation is in reference to chaffinches), which it seems has a relatively similar element duration (~160ms). If these numbers hold, however, this is an interesting discussion point - given similar time to build up pressure and airflow during phonation, why might vocal properties differ between species? I suggest the authors actually report the data first in the results, and then speculate here in the discussion what this might mean.

- Throughout, there is a loose focus on the role of song "in the pair bond", which is an interesting point to make, but somewhat underdeveloped. Indeed, zebra finch females do prefer louder amplitude sounds, though there is a distance component to it. The function of loudness in playbacks as a proxy for distance - which has a clear and stated parallel to pair bonding, which tends to have males vocalizing in very close proximity - would be a beneficial discussion point to speculate upon. This is especially poignant because songs of the zebra finch, and it appears of the star finch as well, are very short-range signals in the wild - and, because the recordings made are of the birds in isolation, whereupon it is unclear whether they are trying to produce "short-range" or "long-range" vocalizations.

Additional comments

It bears repeating that this is a wonderful manuscript, the data of which fills a supremely important gap in the field, and the resources provided will undoubtedly be of great use to the broader community. The authors should be commended on their determination to experimentally typify phonetic relationships that have heretofore been difficult to replicate, and for taking the necessary control that the data they collect can be meaningfully robust before subsequent analyses. I am confident that with the changes I have suggested, and subsequent potential reframing of results and their meaning, this manuscript will fulfill that goal.

Reviewer 2 ·

Basic reporting

This manuscript on vocal constraints of star finches, specifically on their vocal amplitude (a key song feature) in relation to song frequency bandwidth and spectral flatness, is well-written and easy to read. I commend the authors for their work. I only have a few minor issues:
-As complexity is a pretty vague term that can mean many different things (and it is subjective and human-centered, maybe the star finches do not have the same concept of complexity), it is maybe best substituted by simply the measured parameters frequency bandwidth and Wiener entropy.
-Please explain what is meant by nonlinear phenomena as this is not clarified in the manuscript and it is not immediately apparent.
-For line 96-97 the reference is Nemeth et al. 2013 from your reference list, not Nemeth 2012 (I cannot find a phonetogram in Nemeth 2012).
-Currently, the reference list is not entirely in alphabetical order (at least Lampe et al. 2010 is out of place).
-Line 210-211 has the same information content as the sentence following it, I would suggest omiting 210-211 in favour of the following sentence that is very clear in itself.
-Be aware that the dual function of song interpretation of birdsong is being expanded upon by e.g. Rose et al. 2022 and that especially for the zebra finch, another soft singing, non-territorial estrildid and the star finches' best studied close relative, there is some ongoing discussion on whether this song is mainly functioning in the sexual selection context (given that there seems to be not much of a consensus on which males are attractive; Forstmeier & Birkhead 2004; and that females when given the choice prefer the song of their father; reviewed in Riebel 2009; and that most song happens after pair formation; Loning et al. 2023). In zebra finches at least, it is likely that their song plays an important role in individual recognition in social contexts as a name tag (e.g. Bulla & Forstmeier preprint; Loning et al. 2024). Given what this manuscript states at line 201-202, it seems likely that a similar thing could be the case in star finches (very interesting!). I leave it up to the authors whether they are convinced by these arguments and want to slightly reframe their study.
-Are there any studies on the ecology of star finches that mention their short communication distance or non-territoriality? It seems to be an important assumption in the framing, but there are no references for this statement in the text (e.g. at line 101) as the courtship singing and dancing references in line 102 do not address these aspects and are also not on star finches. The recent work on zebra finch amplitude and short communication distances in the wild (Loning et al. 2022) might therefore be relevant to cite at line 101 (and as evidence for estrildid song amplitude differences based on social context, line 142-143).
-Could the authors please expand the discussion with some more general speculation on the implications of the found results on song constraints (e.g. for communication in noise, communication over distance, social organisation)?

References:
Rose, E. M., Prior, N. H., & Ball, G. F. (2022). The singing question: Re-conceptualizing birdsong. Biological Reviews, 97(1), 326–342. https://doi.org/10.1111/brv.12800
Forstmeier, W., & Birkhead, T. R. (2004). Repeatability of mate choice in the zebra finch: consistency within and between females. Animal Behaviour, 68(5), 1017-1028.
Riebel, K. (2009). Song and female mate choice in zebra finches: A review. Advances in the Study of Behavior, 40, 197–238. https://doi.org/10.1016/S0065-3454(09)40006-8
Loning, H., Verkade, L., Griffith, S. C., & Naguib, M. (2023). The social role of song in wild zebra finches. Current Biology, 33, 372–380. https://doi.org/10.1016/j.cub.2022.11.047
Bulla, M. & Forstmeier, W. 2024 No support for honest signalling of male quality in zebra finch song. (preprint at https://doi.org/10.32942/X2D324)
Loning, H., Griffith, S. C., & Naguib, M. (2024). The ecology of zebra finch song and its implications for vocal communication in multi-level societies. Philosophical Transactions B, 379(1905).
Loning, H., Griffith, S. C., & Naguib, M. (2022). Zebra finch song is a very short-range signal in the wild: Evidence from an integrated approach. Behavioral Ecology, 33(1), 37–46. https://doi.org/10.1093/beheco/arab107

Experimental design

Overall, the design seems solid and I have only a few minor things to clarify:
-Line 165-166 what was the reasoning to use songs recorded between 8 and 14h? Did the birds not sing outside of these time windows or was there something else?
-Line 172-173 the difference between a syllable and an element is not completely clear to me, elements are also not named elsewhere much, is the distinction important? an illustration would help here to show how syllables were labeled (and why elements cannot be labeled)
-Line 181-183 does this mean that all the amplitude values reported in this paper are at about 11 cm? This information is important so readers can compare the amplitude of star finches in relation to other species (dB SPL re 20 microPa at a distance of 50 cm or 1 m is common for this) (I am aware the average for your measurements is mentioned on line 105).
-Line 201-202 do the authors mean that star finches have only one song type per individual? with always the same (amount) of syllables? is there minor variation between songs of the same individual as in the zebra finch? Could star finch songs function as name tags as in zebra finches?

Validity of the findings

The study appears to be robust and I am confident that the results reported in this manuscript are valid.

Additional comments

no additional comments

Reviewer 3 ·

Basic reporting

The goal of this study was to identify possible constraints on bird songs, associated with vocal intensity (amplitude), and in a bird species that sings relatively soft songs. As the authors note, studies of vocal amplitude have been relatively rare in the literature, partly because amplitude is hard to measure accurately. I agree that it makes sense to ask how amplitude might be constrained by production mechanics. I also concur that one way to study such constraints is to look at potential trade-offs between amplitude and other song features.

The research team studied the songs of 15 male star finches. Males were recorded individually in a soundproof chamber, and a large sample of syllables (5478) from 20 songs per bird were extracted for acoustic analysis. Five features of song syllables were quantified: amplitude (root mean squared), syllable duration, peak frequency, Wiener entropy, and frequency bandwidth.The amplitude of bird’s songs was calibrated by first recording test tones of known amplitude (as measured by a sound level meter), and then matching the recorded amplitude of those recorded test tones against the recordings of the birds. Distance to the speaker, necessary for calculations of song amplitude, was known for those songs birds sang while on the single perch in the chamber; only those songs were selected for further analysis. The researchers then mapped how the four timing and frequency features mapped onto amplitude (Figures 2 & 3).

These plots, and associated statistical models, illustrate the following patterns: Amplitude varies positively with frequency (Fig 2A); amplitude did not correlate significantly with syllable duration, although it leaned in a positive direction (Fig 2B); and amplitude varied negatively with frequency bandwidth and Wiener entropy (Fig 3). Additional findings were that amplitude tends to decrease as songs progress, and also that males differ in their song amplitudes. A main interpretation focuses on the bandwidth and entropy results (Figure 3); the authors infer that birds singing comparatively loud songs are unable to achieve higher levels complexity. It is interesting to contemplate the ecological and evolutionary consequences of amplitude x complexity trade-offs.

Critiques: Introduction and Interpretation

I had trouble with the argument that we should look for physical constraints in low-amplitude species (abstract, sentence 4). In studies of animal performance, insights into constraints often derive from the study of high-intensity activities (e.g., crushing hard shells, locomotion by burst-swimming or sprinting). As a specific example in bird songs, the study of constraints on trills has focused mainly on songs with rapid repetitions and wide frequency bandwidths, and not on slow songs with narrow bandwidths. Is the argument here that it should be unusually hard to produce very soft sounds? If so, that argument would need some elaboration.

Perhaps the authors are meaning to emphasize the value of studying potential constraints on amplitude in a species that has an unusually wide range of amplitudes. The star finch seems to meet this criteria — I infer from Figures 2 and 3 that this species sings songs that vary across 45 dB, which is notably higher than in the other referenced species from Nemeth et al. 2013 (common blackbirds, reported variation of 26 dB). The observation (Figures 2 & 3) that star finches can sing up to >85 dB suggests that they are not always soft singers.

More importantly, the manuscript lacks specific or clear statements of the study hypotheses, which makes it hard to know what to expect in terms of how the relationships studied should work out. Critically, what is the conceptual basis for the idea that singing at higher amplitudes should limit birds’ capacities in singing complex songs? The explanation given on lines 111-114 doesn’t really say much by itself. By contrast, the expectation for a positive relationship between amplitude and frequency is well-explained (e.g., lines 71-88). I think the explanation for why we might expect positive relationships between amplitude and syllable duration also needs to be expanded (lines 109-111).

Related to this, I”m not sure I’d refer the positive relationships between amplitude and frequency here (and as in Nemeth et al. 2013) as “trade-offs”. Trade-offs usually refer to situations where increases in one feature drive reductions in others.

Experimental design

Critiques: Methods:

The authors state that head direction was controlled by placing a song recorder (with internal microphones) above the center of a perch where the birds sang. I could not figure out how this worked…

I’m worried about the quality of the internal microphones on the Marantz, especially for the measures of amplitude. Typically, researchers measuring amplitude have used high quality omnidirectional microphones. Do we know if those microphones in the Marantz are omnidirectional? If not, even slight variations in the bird’s positions could potentially introduce bias.

The authors note that the test tones were separated from the Marantz by a fixed distance. Was that the same distance as the birds occupied on their perch? If not, was there some correction applied to make sure that the recorded tones could be used accurately as a reference for inferring sound amplitude from birds on the perch?

Critiques: Results:

Given that amplitude is a parameter in all 4 plots, wouldn’t it have made sense to plot amplitude as the independent variable (x-axis) in all four plots? That would make it much easier to visualize how all the plots map onto each other. Relatedly, would it have made sense in the statistical models to treat amplitude as the independent variable? This would fit in more cleanly with the supposition that adjusting for amplitude has secondary effects (e.g., constraints) on other features. Also, I couldn’t figure out why the four plots were split across 2 separate figures. Wouldn’t it have been simpler to include them all in the same figure?

I wonder about the impact of one bird outlier (cyan, I think it’s bird ID sf0020), which shows unusually broad bandwidth and entropy, and is also very soft (Fig 3). To what extent are the trends driven by this one bird, and does it matter for our interpretation?

Critiques: Overall Structure

It would have been helpful to introduce and present the results for the four relationships tested in an order consistent throughout the manuscript (abstract, introduction, methods, and results). As it stands, the order in which the different results were presented was haphazard, which made the narrative harder to follow.

Validity of the findings

Please see comments above

Additional comments

Line by line comments:

Line 21 “the song”, Line 22 “the communication range”, Line 24, “the constraints” and “the limits”; “the” is not needed in all of these phrases. There are additional cases throughout the manuscript (e.g., lines 222, 223).

Line 23, Line 60. I think it’s strange to talk about amplitude of a signal as a signal itself — that confuses what exactly the signal is (is it the signal itself, or its amplitude). I suggest seeking alternative wording.

Line 31, what’s meant by “phonetic structure”? Does this refer to songs’ frequency and timing features?

Line 44, “constrained by predation” I prefer to think of constraints as related to mechanisms (proximate levels), whereas predation has to do with evolution and selection. As a result, this phrase sounded awkward to me. By contrast, I would consider neural costs and body size as potential constraints.

Line 79, the third author’s full last name is “ten Cate”

Line 95, there is an another report of a negative correlation between call duration and amplitude, in White Bellbirds (Podos & Cohn-Haft 2019, Current Biology R1055-1069). The negative correlations in both bellbirds and towhees would likely be explained by another type of constraint, different than the one mentioned on lines 89-91.

Line 97, I revisited Nemeth et al. 2012 and, as far as I can tell, there is no published phonetogram in that paper. Did you mean to refer to the 2013 paper?

Lines 110-111. What is the mechanistic basis for the prediction of a positive relationship between syllable amplitude and duration? Would this really emerge from a production constraint?

Line 138, what is meant by “the song context”?

Line 185, by my count the researchers measured five acoustic parameters.

Line 225, and Line 232; to reiterate, I’m not sure that I’d refer to a positive correlation between amplitude and frequency as a constraint. To me it seems more of a mechanistic coupling, as changes in one should pull the other along in tandem.

Line 252-253, I’d make the same case for the discussion of a relationship between amplitude and syllable duration — as argued, longer song durations would be more a facilitator of achieving higher amplitudes.

Line 254, it might be a technical language issue, but I wouldn’t say that a trade-off provides evidence “in addition to” physical constraints. Rather, that would be evidence itself for physical constraints.

Lines 261-263, this comes closer than any previous language in the paper regarding the central hypothesis being tested, as to why higher amplitude sounds might be limited in their complexity. I suggest moving these statements to the introduction, stating them as hypotheses, and expanding on them. What would it take to show that loud songs are limited to the frequencies that are amplified most effectively by a bird’s vocal tract? Why shouldn’t they be able to sing loud songs outside of these frequencies?

Lines 270-271, this seems like an overstatement to me. At least some of the other studies of vocal amplitude cited have speculated on the mechanics of sound production, and thus considered potential constraints, even if not stated as such. Do you mean to say that yours if the first study to look at trade-offs with amplitude, in which louder songs experience changes in other song features? I’ll also point out this paper: Jakobsen, L., et al. (2021), How loud can you go? Physical and physiological constraints to producing high sound pressures in animal vocalizations. Front. Ecol. Evol. 9: 657254

Line 281-282, the point is made here (and at several other points) that star finches show unusually high vocal complexity, which could make them an especially good species for studies of vocal constraints. However, I’m not convinced the authors made this case. The measures here for complexity (frequency bandwidth, Weiner entropy) are presented just for star finches — is it really the case that these values are notably broader than in other species? Also, this point also doesn’t map on directly to the prior argument from the introduction for looking at star finches, which is that they show unusually wide variation in song amplitude (I also wonder if this is the case— again some sort of comparisons would be helpful, perhaps to sparrows and warblers that are known to sing both regular songs and soft songs).

---

## Round 0.2 · Minor Revisions

· Academic Editor

Minor Revisions

Thank you for revising your manuscript. There are some minor comments that two reviewers suggested should be addressed before submitting the revised manuscript.

Reviewer 1 ·

Basic reporting

no comment

Experimental design

no comment

Validity of the findings

no comment

Additional comments

I am grateful to the authors for their careful reading and implementation of my feedback. I see no reason why this paper should not be published as is. I feel wholeheartedly they have conceptually engaged with my critiques, and as a result their manuscript is now much stronger. I look forward to its publication, and their follow-up work.

Reviewer 2 ·

Basic reporting

The manuscript is much improved - I just have some very minor things left:

Ln 115: something wrong on this line with the reference after singing - please check your reference manager software.

Line 135: I still don't find the explanation of nonlinear phenomena clear (and how it is distinct from biphonation). (What is two-voice biphonation? This is more obvious but to spell it out would not hurt.) How are these intricate modulations produced?

Line 237: Do the authors mean the model was ran without this individual? I suspect that is the case but it is not 100% clear what was done.

Line 257: Do the authors mean to refer to figure 2C instead of 2A? This sentence needs extra clarification. Do the authors mean that the same model was ran without the inclusion of this individual?

Line 318 add a .

Figure 1: in the gray boxes of the figure I see 'freqency' and 'freuqncy', these should all be 'frequency'.

Experimental design

If the authors could clarify the nonlinear phenomena and biphonation that I already mentioned at point 1 for the prediction, as well as clarify the seperate models that are mentioned for a few individuals, I have no further comments here.

Validity of the findings

The authors have convinced me of the validity of the findings with this rigorous study that I think will be an excellent addition to our current understanding of vocal amplitude constraints.

The last sentence 'When liberated from the ecological demands of signal transmission over long distances, birds can use soft yet complex syllables that function within the pair bond.'
does not follow directly from the results. Maybe tone it down a little bit or make it more clear that this is speculative. E.g.:
'It is likely that the absence of the ecological demands of signal transmission over long distances lead to their soft yet complex syllables, which likely function in short-range contexts, e.g. within the pair or at assemblages.'

Additional comments

For Figure 1 it could be interesting to also add the male SF00020 as that was the male you tentatitvely excluded just to check if its inclusion mattered, because this male seemed a bit different in terms of bandwidth and amplitude.

Reviewer 3 ·

Basic reporting

I was reviewer #3 on the prior round. I read the revised manuscript fresh, and then looked over the response letter and explanation of changes made in response to the prior round of review.

The manuscript was much easier for me to follow this time around. In my view, the authors solved virtually all of the queries raised. This is a nice data set and I think will be of broad interest. It is indeed fascinating to consider how amplitude might shape the production of other bird song features.

There is however a terminology issue that kept coming up for me, which I continue to find confusing (and I hope that I can explain the issue with greater clarity in this second round).

The data presented in Figure 2C and D show a negative relationship and are consistent with a trade-off. As birds sing more loudly, it seems they are stuck simplifying their songs. Or, as birds sing more complex, it seems they are stuck singing more softly. Whichever explanation, I support the inference that these relationships are shaped by "constraints". This term nicely denotes the difficulties birds will have in potentially trying to maximize both parameters at the same time (and I presume they are "constrained" from singing songs that are both loud and complex.)

But, the data presented in Figure 2A shows a positive relationship: As birds sing more loudly, their vocal frequencies go up; or, as they sing at higher frequencies, their songs get louder. There is no visible trade-off here in the data shown -- rather, increases in one parameter seem to drive increases in the other, for mechanistic reasons. The key point here is that, at least in my view, this mechanistic relationship is not really a "constraint", insofar as increasing one enables (or provides opportunities for) increases in the other, rather than restricting the other.

I recognize that my definition of what constitutes a constraint might be more limited and focused than the one the authors adopt. I believe that authors are free to use terminology as they like. And perhaps the authors mean to argue that the birds are indeed constrained in these axes, from singing songs that are low frequency and high amplitude, or high frequency and low amplitude. But I don't think that such conclusions can be inferred from the data actually shown. The focus of the data is on how the parameters enhance one another.

I thought I'd share my perspective because I see only benefits (in increased clarity) in using the more restricted definition for "constraints" (which, after all, maps more directly onto the conventional common use of the word).

Two other main comments:

-- I am really having trouble, in this cleaner revision, in understanding the first sentence of the conclusion section. In my reading, prior research on vocal constraints has in fact focused closely on phonetic constraints (which I presume refers to the syllable- and song-level acoustic structure, such as timing versus frequency traits), whereas those other variables mentioned (social aggression, predation, neural costs, body size) have been studied secondarily, as having potential impacts on how acoustic phonology is constrained.

-- for Figure 2, I for one am unable to match the colors of the points to the key with the individual IDs. My color discernment is just not that precise.

Experimental design

no concerns

Validity of the findings

no concerns

Additional comments

no concerns

---

## Round 0.3 · accepted · Accept

· Academic Editor

Accept

Thank you for submitting the revised manuscript, which addresses the minor comment suggested by one of the two reviewers. Congratulations, and thank you for submitting your manuscript to PeerJ.